# Regulation Effectiveness and Mechanism of Biotransformation Pathway on the Toxicity of Microcystin-LR Target to Protein Phosphatase 2A

**DOI:** 10.3390/ijerph20020964

**Published:** 2023-01-05

**Authors:** Huiqun Yu, Jiyuan Cui, Yixue Xu, Li-Juan Feng, Wansong Zong

**Affiliations:** College of Geography and Environment, Shandong Normal University, 88# East Wenhua Road, Jinan 250014, China

**Keywords:** microcystin-LR, protein phosphatase 2A, toxicity regulation, biotransformation pathway

## Abstract

Biotransformation is recognized as a potential pathway to regulate the environmental risk of microcystins (MCs). To explore the regulation effectiveness and mechanism of the biotransformation pathway, six typical MCLR-biotransformation products (MCLR-BTPs) were prepared, and their inhibition effects on protein phosphatase 2A (PP2A) were evaluated. The inhibition effects of the MCLR-BTPs generally decreased with the increase in biothiol molecular weights and polarity, indicating that biotransformation was an effective pathway through which to regulate MCLR toxicity. To further explore the regulation mechanism, the key interaction processes between the MCLR/MCLR-BTPs and the PP2A were explored by homology modeling and molecular docking. The introduced biothiols blocked the covalent binding of Mdha^7^ to Cys_269_ but strengthened the hydrogen bond “Mdha^7^”→Arg_268_. The changed “Mdha^7^” intervened the combination of MCLR-BTPs to PP2A by weakening the hydrogen bonds Arg^4^←Arg_214_, Arg^4^→Pro_213_, Adda^5^←His_118_, and Ala^1^←Arg_268_, and the ionic bond Glu^6^-Mn_1_^2+^. The weakening combination of the MCLR-BTPs to PP2A further attenuated the interactions between the conserved domain and the Mn^2+^ ions (including the ionic bonds Asp_57_-Mn_1_^2+^ and Asp_85_-Mn_1_^2+^ and the metal bonds Asp_57_-Mn_1_^2+^ and Asn_117_-Mn_1_^2+^) and increased the exposure of the Mn^2+^ ions. Meanwhile, the weakened hydrogen bond Arg^4^←Arg_214_ facilitated the combination of the phosphate group to Arg_214_ (with increased exposure). In this way, the catalytic activity of the PP2A was restored.

## 1. Introduction

Microcystins (MCs), the most common cyanotoxins detected in toxic cyanobacterial blooms, are a potential health risk to aquatic organisms and animals, including humans [1,2]. Human exposure to MCs occurs by inhalation, dermal exposure, and mainly by oral consumption of contaminated food or drinking water [3,4]. Protein phosphatases (PPs) play a pivotal role in regulating reversible phosphorylation in many intracellular signaling pathways [5]. After the uptake of MCs in humans, MCs mainly irreversibly inhibit the activity of PPs, resulting in hyperphosphorylation of vital cellular proteins, the disintegration of hepatocyte structures, apoptosis, intrahepatic hemorrhage, and death [6,7]. A crystal structure analysis of the complexes of MCLR (the most widespread and toxic MC) and PPs showed that MCs undergo two-step interactions with the catalytic centers of PPs [8,9]. In the first step, the side chain of Adda^5^ was rapidly wrapped in the hydrophobic cage structure of the PPs. In the second step, the unsaturated carbonyl of Mdha^7^ is irreversibly bound to specific cysteine residues by nucleophilic addition reaction to form covalent bonds. As a result, the key interactions between the conserved domains of PPs (especially nine strictly conserved amino acids) and the metal ions/introduced phosphate group changed accordingly, resulting in the inhibition of the PPs’ activity [10].

In view of the toxicity of MCs, controls on their environmental risk are of great importance. Significant advances in water-treatment technology have produced some solutions, including micro-biological degradation, disinfection, and active carbon adsorption [11,12,13,14]. In addition to these strategies, the biotransformation pathway is considered a valuable strategy through which to regulate the toxicity of MCs [15,16,17]. Pioneering toxicology studies found that the unsaturated Mdha^7^ of MCs can covalently bind to dissociative biothiols (forming MC-biotransformation products, MC-BTPs) and block the covalent binding of MCs to PPs [18,19]. However, partial studies proposed that the first step was required for the inactivation of PPs activity, whereas the formation of covalent adducts was not [20]. In addition, partial studies found potential inhibition effects of MC-BTPs on PPs [21,22]. On account of this, the regulation effectiveness of the biotransformation pathway needs to be fully evaluated, and the molecular mechanism of the biotransformation pathway needs to be further explored.

To evaluate the regulation effectiveness of the biotransformation pathway, the first problem to be tackled is to prepare sufficient MC-BTPs. As MC-BTPs are the nucleophilic addition products of MCs and biothiols, they can be conveniently prepared in vitro by simulating the addition reaction of MCs and related thiols [18]. According to the traditional separation and purification methods of MCs [23,24], MC-BTPs can be further identified, purified, and used for subsequent toxicity evaluation. Unfortunately, limited by the crystal structures of MC-BTP-PP complexes, the molecular mechanism of the biotransformation pathway is difficult to elucidate. A pioneering study by Xu et al. provided a new perspective on evaluating the interactions between toxins and proteins without corresponding crystal structures [25]. They constructed the interaction models for typical MC-PP2A complexes and explored the key interactions based on homology modeling and molecular docking. Based on the same strategy, the interaction models for MC-BTP-PP complexes can be constructed, and the key interactions between MC-BTPs and PPs can be obtained.

Several MCLR-BTPs were prepared by simulating the nucleophilic addition reactions of MCLR with six typical biothiols (see Figure 1) and identified by MS and MS/MS. After chromatographic preparation and purification, the inhibition effects of MCLR and MCLR-BTPs on PP2A were evaluated by a traditional PP inhibition assay. With the assistance of homology modeling and molecular docking, the models of the MCLR-BTPs and PP2A were constructed, and the candidate interaction parameters, such as interaction areas and related chemical bonds, were obtained. Taking the inhibition data and candidate interaction parameters into consideration, the key interaction parameters were screened, and the crucial interaction processes were explored. In this way, the regulation effectiveness and the mechanism of the biotransformation pathway on the toxicity of the MCLR target to PP2A were elucidated in detail. The current study draws increased attention to the secondary environment risks caused by MC-BTPs, thus contributing to the improvement of regulatory strategies against MC/MC-BTP biotoxicity and minimizing their impact on human health. This study not only contributes to a comprehensive understanding of the toxicity of MC-BTPs but also helps to achieve the regulation of MC toxicity through exogenous biothiols. Therefore, it has important theoretical and practical application value.

## 2. Materials and Methods

### 2.1. Materials

The MCLR standard was purchased from Sigma (Saint-Quentin Fallavier, France). The glutathione (GSH), Cys, Cys-Gly, 2-hydroxy-1-ethanethiol (βME), thiolacetic acid (AcSH), homocysteine (Hcy), bovine serum albumin (BSA), p-nitrophenyl disodium orthophosphate (p-NPP), and tris(hydroxymethyl)aminomethane (Tris), dithiothreitol, and MnCl_2_ were purchased from Sinopharm (Shanghai, China). A total of 1 unit (U) is the amount of enzyme that catalyzes the reaction of 1 µmol of substrate per minute under standard conditions. PP2A activity is quoted in units per ml (U/mL). The PP2A (1200 U/mL) from rabbit skeletal muscle was obtained from EMD Millipore (Darmstadt, Germany). The HPLC-grade acetonitrile, trifluoroacetic acid, and methanol were obtained from Merck (Darmstadt, Germany).

### 2.2. Addition Reaction of Biothiols to MCLR

The MCLR-BTPs were prepared by mixing 2 μM MCLR and 500 μM GSH (Cys, Cys-Gly, βME, AcSH, or Hcy) in 5% K_2_CO_3_ and incubating at room temperature for 2 h [18]. Subsequently, the reaction products were neutralized with 0.2 M HCl and applied to conditioned Cleanert C_18_ SPE cartridges (500 mg, Bonna-Agela) that were washed with 10 mL 100% methanol and 15 mL distilled water. The impurities and MCLR-BTPs were eluted with 10 mL 10% acetonitrile and 10 mL 80% acetonitrile, respectively. The eluted samples were evaporated to dryness in N_2_ flow and resuspended in 1 mL acetonitrile. The eluted samples were stored at −20 °C.

### 2.3. Identification and Preparation of MCLR-BTPs

#### 2.3.1. MS and MS/MS Analysis of MCLR-BTPs

The crude extracts of MCLR-BTPs were mixed with isometric acetonitrile (containing 0.1% trifluoroacetic acid), after which they were injected into a maXis UHR-TOF mass spectrometer for preliminary identification. Typical MS parameters were set as follows: positive-ion-spray ionization pattern, source voltage 4 kV, cone voltage 0.5 kV, desolvation gas (N_2_) 0.4 bar, dry-gas (N_2_) heater 180 °C, dry-gas-flow rate 4 L/min, full scan *m/z* 300–1500. The structures of MCLR-BTPs were further identified by MS/MS. Based on MS/MS analysis, the prepared MCLR-BTPs were accurately identified by comparing their secondary structures with that of MCLR. MS/MS parameters were set as in MS analysis, except that collision gas N_2_ was used, and collision energies were adjusted at 45–55 eV.

#### 2.3.2. Preparation of MCLR-BTPs

The separation of MCLR-BTPs was performed through an Agilent SB-C_18_ column (9.4 × 250 mm, 5 μm) with 10 mL methanol and 10 mL high-purity water. Water and acetonitrile were used as mobile phase A and mobile phase B, both containing 0.1% trifluoroacetic acid. The gradient elution was programmed as follows: 0–5 min, 20% B; 5–35 min, 20%–80% B; 35–40 min, 80% B; 40.1–45 min, 20% B. The column temperature and the flow rate were set to 35 °C and 2 mL/min, respectively. After chromatographic separation, each MCLR-BTP was detected by maXis UHR-TOF mass spectrometer. The specific MS/MS parameters were set as in Section 2.3.1. The purified MCLR-BTPs were collected at their specific retention times. Purified samples for MCLR-BTPs were dried with N_2_ and dissolved in 100 μL acetonitrile. The MS analysis of prepared MCLR-BTPs used the MCLR standard as a reference.

### 2.4. PP2A-Inhibition Assay for MCLR and MCLR-BTPs

Using a colorimetric protein-phosphatase-inhibition assay (mainly referring to the experimental method of Zong et al. [26]), the inhibition effects of MCLR/MCLR-BTPs on PP2A were evaluated by measuring PP2A activity. First, PP2A was diluted to 2 U/mL with buffer solution (50 mM tris-HCl, 2 mM dithiothreitol, 1 mM MnCl_2_) and 1 g/L BSA. Next, 10-microliter PP2A and 100-microliter samples were mixed in a 96-well polystyrene microplate, and the microplate was gently shaken for 15 min. After shaking slightly, 90 μL p-NPP was added to the microtiter plate and incubated at room temperature (25.0 °C) for 15 min. Finally, the absorbance ODS_405_ of incubated samples was measured with Thermo/max microplate reader after 1 h. Three parallel trials were run for each experiment. The inhibition effect of target samples on PP2A was expressed as follows: I_PP2A_ (%) = (A_toxins_ − A_blank_)/(A_control_ − A_blank_) × 100%. In the control group, MCLR/MCLR-BTPs were replaced by distilled water, and in the blank group, MCLR/MCLR-BTPs and PP2A were replaced by distilled water.

### 2.5. Molecular Simulation for the Interactions between MCLR/MCLR-BTPs and PP2A

Molecular-docking simulation was performed by Molecular Operating Environment software (MOE, version 20.09, Cloud Scientific, Shanghai, China). The experimental steps were as follows: firstly, the original model for the MCLR-PP2A complex was downloaded from Protein Data Bank (PDB code 2IE3). When the model for the MCLR-PP2A complex was introduced into MOE, PP2A was optimized by “building missing loops” and adjusting the charges of the whole system [27]. Secondly, the interaction models for MCLR-BTP and PP2A complexes were preliminary constructed by means of homology modeling: the original ligand MCLR in the optimized model of MCLR-PP2A was replaced by identified MCLR-BTPs. The models of MCLR-BTPs and PP2A were minimized for energy optimization [27]. Next, the interactions between toxins and PP2A were simulated by molecular docking. To ensure the comparability of MCLR-BTPs with their original toxin, “template dock” mode was adopted (in “template dock” mode, the placement and refinement of MCLR-BTPs were in keeping with their original toxin). The specific docking parameters were set as follows: amber 10 EHT, solvation r-field, temperature 25.0 °C, pH 7.4, salinity 0.05 M [25]. Finally, the candidate interaction parameters (combination areas, related surface areas, hydrogen bonds, metal bonds, ionic bonds, exposure areas of Mn^2+^ ions, and introduced phosphate group) related to the combination of toxins to PP2A could be obtained.

### 2.6. Statistical Analysis

All inhibition data obtained were expressed as the mean ± standard error of three experiments carried out independently (n = 3). The correlations between inhibition data and candidate interaction parameters were analyzed by IBM SPSS Statistics (version 26.0, Chicago, IL, USA). Significance levels were classified as extremely significant (*p* < 0.01), significant (*p* < 0.05), or not significant (*p* > 0.05). One-way analysis of variance (ANOVA) followed by LSD post hoc tests was used to check significant differences between groups. Values of *p* < 0.05 were considered as having statistical significance.

## 3. Results and Discussion

### 3.1. MCLR-BTP Identification and Preparation

Conjugated with biothiols, MCLR was converted into specific MCLR-BTPs with different molecular weights and detected by a mass spectrometer. For MCLR (C_49_H_74_N_10_O_12_), the MS signal presented at *m/z* 995.5557 should correspond to its single-proton product (Figure 2A). In the selected electrophilic addition sample, MCLR remained but had a lower intensity than the newly formed ion, with an MS signal at *m/z* 1073.6902 (Figure 2B). Since the molecular weight of the βME was about 78.1345 Da, the MS signal was assigned to the addition product of the βME to the MCLR. For the other electrophilic addition samples, five newly formed MS signals for the single-proton MCLR-BTPs were detected at *m/z* 1071.6745, 1116.7152, 1130.7417, 1302.8794, and 1173.7664 (Appendix A). As the molecular weights of the AcSH, Cys, Hcy, GSH, and Cys-Gly were about 76.1188 Da, 121.1595 Da, 135.1860 Da, 307.3237 Da, and 178.2107 Da, respectively, these newly formed MS signals were assigned to the conjugation products of the biothiols to MCLR.

With the assistance of the MS/MS analysis, the MCLR-BTPs were further identified by comparing their secondary structures with that of MCLR. Partial basic fragment ions of the MCLR were detected at *m/z* 135.0803, 213.0832, 286.1477, 397.2079, 470.3124, 553.3072, 599.3542, 682.3957, and 866.5149 (Figure 2C), corresponding to the secondary structures of [PhCH_2_CH(OCH_3_)]^+^, [Glu-Mdha+H]^+^, [MeAsp-Arg+H]^+^, [Mdha-Ala-Leu-MeAsp+H]^+^/[Glu-Mdha-Ala-Leu+H]^+^, [Arg-Adda+H]^+^, [Mdha-Ala-Leu-MeAsp-Arg+H]^+^, [MeAsp-Arg-Adda+H]^+^/[Arg-Adda-Glu+H]^+^, [Arg-Adda-Glu-Mdha+H]^+^, and [Mdha-Ala-Leu-MeAsp-Arg-Adda+H]^+^/[Arg-Adda-Glu-Mdha-Ala-Leu+H]^+^ [14]. To use an example, the MCLR-βME not only had several identical fragment ions to the MCLR (e.g., 135.0803, 286.1476, and 470.3123) but also had several new fragment ions at *m/z* 291.2175, 475.3422, 631.4416, 760.5304, and 944.6493 (Figure 2D), corresponding to the ions of [Glu-Mdha+H]^+^ + 78.1343, [Mdha-Ala-Leu-MeAsp+H]^+^/[Glu-Mdha-Ala-Leu+H]^+^ + 78.1343, [Mdha-Ala-Leu-MeAsp-Arg+H]^+^ + 78.1344, [Arg-Adda-Glu-Mdha+H]^+^ + 78.1347, and [Mdha-Ala-Leu-MeAsp-Arg-Adda+H]^+^/[Arg-Adda-Glu-Mdha-Ala-Leu+H]^+^ + 78.1344. Based on the same strategy, other MCLR-BTPs were identified (see Appendix A). By comparing these fragment ions, it was found that the mass changes were all related to the Mdha^7^ residue. On these grounds, the MCLR-BTPs described above should be formed by the nucleophilic addition reaction of biothiols (βME, AcSH, Cys, Hcy, GSH, and Cys-Gly) to the unsaturated carbonyl groups in Mdha^7^ residue.

### 3.2. Evaluation of the Inhibition Effects of MCLR/MCLR-BTPs Targeted to PP2A

To explore the regulation effectiveness of the biotransformation pathway, the MCLR-BTPs in the crude extract were purified with preparative-chromatography techniques [14]. The preparation and purification information for the MCLR-BTPs is listed in Appendix A. As the prepared samples had higher concentrations (>1000 μmol/L) and higher purity (>96.7%), they were directly used in the PP2A inhibition assay.

Figure 3 shows that the MCLR-BTPs all inhibited PP2A activity, although these inhibition effects were weaker than with the MCLR. As the toxin concentration increased, the decrease in PP2A activity became more obvious and occurred in dose–effect relationships. The inhibition effects of the MCLR and MCLR-BTPs on the PP2A were significantly different between the groups (*p* < 0.05). At 1 nM, the inhibition effects of the toxins were divided into five categories (a MCLR; b MCLR-βME; c MCLR-AcSH; d MCLR-Cys, MCLR-Hcy; e MCLR-GSH and MCLR-Cys-Gly) according to the ANOVA. Based on the same methodology, the inhibition effects of the toxins were divided into four categories at 10 nM, while the inhibition effects of the toxins were divided into six categories at 100 nM. To summarize, the inhibition sequence was in the decreasing order of MCLR > MCLR-βME > MCLR-AcSH > MCLR-Cys > MCLR-Hcy > MCLR-GSH > MCLR-Cys-Gly. Combined with the basic properties of the biothiols, it was found that the inhibition effects of the MCLR-BTPs on the PP2A were correlated with the polarity and molecular weights of the biothiols (|R| > 0.757, *p* < 0.049). On these grounds, it was speculated that the polarity and steric effect of “Mdha^7^” could affect the combination of MCLR-BTPs and PP2A, thus playing a role in regulating toxicity. However, what needs to be emphasized are the inhibition effects of the MCLR-BTPs (especially MCLR-βME and MCLR-AcSH) on the PP2A.

Previous reports confirmed the important role of biothiols in regulating the biotoxicity of MCLR [28,29]. Our findings are consistent with those of previous studies. According to our results, the toxicity of the MCLR-BTPs was decreased to varying degrees compared with the native toxin, indicating that biotransformation was an effective pathway to regulate the toxicity of MCLR. In addition, a previous study showed that MCLR-BTPs still feature potential toxicity [22]; our findings were consistent with this study and suggested that the potential toxicity of the MCLR-BTPs might be related to the polarity and steric effect of “Mdha^7^”. In conclusion, the biothiols were critically important for the detoxification of the MCLR, but the secondary environmental risk caused by the MCLR-BTPs was non-negligible and deserves further attention.

### 3.3. Simulation of the Interactions between MCLR-BTPs and PP2A Based on Homology Modeling and Molecular Simulation

The biothiols had an evident regulation effect on the toxicity of the MCLR, but the molecular mechanism of the biotransformation pathway was difficult to study thoroughly because the crystal structures of the MCLR-BTP-PP2A complexes were limited. As the structures of the MCLR-BTPs were similar to those of the protoxins, homology modeling was used to construct the interaction models of MCLR-BTPs and PP2A (Figure 4). The crystal structure for the MCLR-PP2A complex was obtained from the Protein Data Bank (PDB ID: 2IE3), after which the protein structure was preprocessed after introducing the MCLR-PP2A complex into the MOE software. Based on the model for the MCLR-PP2A complex, the models for the MCLR-BTPs and PP2A were constructed through homology modeling, which filled the gap in the interaction models for MC-BTP-PP complexes, contributing to the extraction of the intermolecular interactions. On this basis, 85 candidate interaction parameters (combination areas, related surface areas, related chemical bonds) between toxins and PP2A were obtained using the MOE dock (listed in Appendix A).

### 3.4. Pearson Correlation Analysis for the Candidate Interaction Parameters and Inhibition Data

The correlations between the inhibition data and the candidate interaction parameters were identified by Pearson correlation analysis. Regression analysis was not used to avoid deleting the valid parameters associated with a few limited amino acid residues. Correlation analysis found that the candidate interaction parameters exhibited diversified correlations with the inhibition data at different toxin levels (Figure 5 and Appendix A): in total, 34 interaction parameters were positively correlated with the inhibition data at all or partial concentrations, while 51 interaction parameters were negatively correlated with the inhibition data at all or partial concentrations. Of the interaction parameters, 10 interaction parameters were extremely significantly correlated with the inhibition data (*p* < 0.01), while 21 interaction parameters were significantly correlated with the inhibition data (*p* < 0.05). The candidate interaction parameters that were significantly or extremely significantly correlated with the inhibition data (*p* < 0.05 or *p* < 0.01) were considered to be the key interaction parameters. The key interaction parameters were important to explore the interaction processes between MCLR/MCLR-BTPs and PP2A and helped to clarify the molecular mechanism of the biotransformation pathway.

Therefore, Venn diagrams were further used to screen the key interaction parameters (Figure 6A,B). At the level of *p* < 0.01, the combination areas for Ala^1^→PP2A, Adda^5^→PP2A and “Mdha^7^”→PP2A, the positive accessible surface area for Glu^6^→PP2A, the polar surface area for “Mdha^7^”→PP2A, the hydrogen bonds for Ala^1^←Arg_268_, Adda^5^←His_118_, and the amino acid associated with the binding of the phosphate group for Arg_214_ had a highly significant correlation with the toxin toxicity at the three tested concentrations. The combination area for Arg^4^→PP2A and the negative accessible surface area for the toxin to PP2A were significantly correlated with the toxin toxicity at 1 nM. At the level of *p* < 0.05, the combination area for the toxin to the PP2A, the positive accessible surface areas for the toxin to the PP2A, “Mdha^7^”→PP2A, the hydrophobic surface area for Adda^5^→PP2A, the polar surface areas for the toxin to the PP2A, MeAsp^3^→PP2A, the hydrogen bonds for Arg^4^←Arg_214_, “Mdha^7^”→Arg_268_, the ionic bonds for Glu^6^-Mn_1_^2+^, Asp_57_-Mn_1_^2+^, Asp_85_-Mn_1_^2+^, the metal bonds for Asp_57_-Mn_1_^2+^, Asn_117_-Mn_1_^2+^, and the catalytic center exposure areas for Asp_57_ + Mn_1_^2+^ and Asp_57_ + Asp_85_ + Asn_117_ + His_167_ + His_241_ + Mn_1_^2+^ were highly and significantly correlated with toxin toxicity at the three tested concentrations. The hydrogen bond for Arg^4^→Pro_213_ was highly and significantly correlated with the toxin toxicity at 1 nM and 10 nM. The combination area for Arg^4^ → PP2A and the negative accessible surface area for “Mdha^7^”→PP2A were highly and significantly correlated with the toxin toxicity at 10 nM and 100 nM. The combination area for MeAsp^3^→PP2A, the negative accessible surface area for the toxin to the PP2A, and the polar surface area for Arg^4^→PP2A were significantly correlated with the toxin toxicity at 1 nM. The catalytic center exposure area for Asn_117_ + Mn_1_^2+^ was significantly correlated with the toxin toxicity at 10 nM. To conclude, the interaction parameters that were significantly correlated with the toxin toxicity at two and three tested concentrations were crucial for elucidating the regulation mechanism of the biotransformation pathway.

The key interaction sites were identified by the statistical analysis of key interaction parameters (based on the structural units of the toxins, Mn^2+^ ions in the catalytic center, and the phosphate group) (Figure 6C,D). A statistical-frequency analysis (Figure 6C) manifested eight key interaction parameters related to the Mn_1_^2+^ ion, five key interaction parameters related to “Mdha^7^”, four key interaction parameters related to Arg^4^, three key interaction parameters related to Adda^5^, two key interaction parameters related to Ala^1^/Glu^6^/MeAsp^3^, and one key interaction parameter related to the phosphate group. Combined with the statistical analysis of the total |$\bar{R}$| values related to the sites listed above (Figure 6D), the interaction sites all participated in the combination of the MCLR-BTPs to PP2A, and their contributions were in the following sequence: Mn_1_^2+^ ion >“Mdha^7^” > Arg^4^ > Adda^5^ > Ala^1^ > Glu^6^ > MeAsp^3^ > phosphate group. In particular, the Mn_1_^2+^ ion/“Mdha^7^” had a prominent influence on the combination of MCLR/MCLR-BTPs to PP2A, Arg^4^/Adda^5^ had a considerable influence on the combination of MCLR/MCLR-BTPs to PP2A, while Ala^1^/Glu^6^/MeAsp^3^/phosphate group had a certain influence on the combination of MCLR/MCLR-BTPs to PP2A.

Since the Mn^2+^ ions in the catalytic center were important to maintain PPs’ activity [30], the key interactions involving the Mn_1_^2+^ ion in the biotransformation pathway were critical for the recovery of the PP2A’s catalytic activity. This finding agreed with previous studies, in which the inhibition effects of toxins on PP2A were primarily mediated by Mn_1_^2+^ ions [25]. Among the structural units of the MCLR/MCLR-BTPs, “Mdha^7^” was more significant than that of other sites. Therefore, it was speculated that the change in the “Mdha^7^” in the biotransformation pathway was crucial to the regulation of MCLR toxicity. This finding was consistent with the conclusion of the previous inhibition assay, in which it was found that the polarity and steric effect of “Mdha^7^” might play a crucial role in affecting the combination of MCLR-BTPs to PP2A. This confirmed the role of “Mdha^7^”, which may play an essential role in restraining the combination of MCLR-BTPs to PP2A and in protecting human health against MCLR toxicity.

### 3.5. Molecular Mechanism behind Biothiols’ Regulation of the Inhibition Effects of MCLR-BTPs on PP2A

With the assistance of a two-dimensional ligand–receptor interaction diagram, the key interaction sites and related key interactions were graphically illustrated (Figure 7). The regulation mechanism of the biotransformation pathway can be elucidated as follows.

When the biothiols were introduced, the covalent binding of toxins to Cys_269_ in the catalytic center was directly disrupted; however, meanwhile, the hydrogen bond “Mdha^7^”→Arg_268_ was directly strengthened. The increased combination areas of “Mdha^7^” to PP2A can be attributed to the strengthened hydrogen bond described above and the extra areas of the biothiols. The introduced polar biothiols strengthened the electropositive and electronegative interactions between “Mdha^7^” and PP2A; thus, the positive accessible surface areas, negative accessible surface areas, and polar surface areas of “Mdha^7^” to the PP2A increased accordingly. Next, the structural changes in the “Mdha^7^” and the steric effect of the introduced biothiols intervened in the combination of the adjacent residues to the PP2A by weakening the hydrogen bonds Arg^4^←Arg_214_, Arg^4^→Pro_213_, Adda^5^←His_118_, and Ala^1^←Arg_268_, and the ionic bond Glu^6^-Mn_1_^2+^. Correspondingly, the combination areas of Arg^4^, Adda^5^, and Ala^1^ to PP2A and the positive accessible surface area of Glu^6^ to PP2A generally decreased. Meanwhile, the structural changes in the “Mdha^7^” and the steric effect of the introduced biothiols strengthened the interactions between MeAsp^3^/Arg^4^ and the polar amino acids but weakened the hydrophobic interactions between Adda^5^ and the PP2A, resulting in an increase in polar surface areas and a decrease in hydrophobic surface areas, respectively. Subsequently, the weakening combination of MCLR-BTPs to PP2A further attenuated the interactions between the partial key amino acids in the conserved domain and the Mn^2+^ ions, including the ionic bonds Asp_57_-Mn_1_^2+^ and Asp_85_-Mn_1_^2+^ and the metal bonds Asp_57_-Mn_1_^2+^ and Asn_117_-Mn_1_^2+^. The interactions associated with the Mn^2+^ ions were weakened, resulting in the increased exposure areas of the Mn^2+^ ions. At the same time, the weakening of the hydrogen bond Arg^4^←Arg_214_ increased the exposure of the Arg_214_; thus, the combination of the phosphate group to Arg_214_ was facilitated. Both the exposure of the Mn^2+^ ions and the combination of the phosphate group to Arg_214_ increased, resulting in the restoration of the catalytic activity of the PP2A.

A previous study on PPs found that the catalytic core of PP2A contained nine strictly conserved amino acids, of which six (His_59_, His_167_, His_241_, Asp_57_, Asp_85_, and Asn_117_) coordinate with Mn^2+^ ions and three (Arg_89_, His_118_, Arg_214_) bind to the phosphate group [10]. In our study, the changed interactions between six conserved amino acids and Mn^2+^ ions interfered with the exposure of Mn^2+^ ions, which mainly involved the exposure of the Mn_1_^2+^ ions coordinated with Asp_57_, Asp_85_, and Asn_117_. Therefore, the increased exposure areas of the Mn_1_^2+^ ions coordinated with Asp_57_/Asp_85_/Asn_117_ were crucial for the recovery of the PP2A’s catalytic activity. In addition, the changed interactions between three conserved amino acids and the phosphate group interfered with the binding of the phosphate group to the three conserved amino acids. The binding of Arg_214_ to the phosphate group was obviously increased (with a weakened hydrogen bond, Arg^4^←Arg_214_), while the binding of Arg_89_/His_118_ to the phosphate group was not significantly affected; thus, Arg_214_ was considered to have exerted an important effect on the molecular mechanism of the biotransformation pathway.

The analysis of the molecular mechanism contributed to the comprehensive evaluation of the toxicity of the MCLR-BTPs and to regulating the potential threat of MCLR-BTPs, thereby reducing the secondary environmental risk of MCLR-BTPs and minimizing their impacts on human health (preventing hyperphosphorylation of important cellular proteins).

## 4. Conclusions

This research investigated the regulation effectiveness and mechanism of the biotransformation pathway on the toxicity of the MCLR target to PP2A. Based on the preparation and purification of six typical MCLR-BTPs, the regulation effectiveness of the biotransformation pathway was fully verified through a PP2A inhibition assay. The results showed that biotransformation was an effective pathway through which to regulate the toxicity of MCLR. However, the secondary environment risks of the MCLR-BTPs were non-negligible. With the assistance of a molecular simulation, the specific regulatory mechanism of the biotransformation pathway can be elucidated as follows. The introduced biothiols directly blocked the covalent binding of “Mdha^7^” to PP2A and directly strengthened the hydrogen bond “Mdha^7^”→Arg_268_. Next, the changed “Mdha^7^” intervened in the combination of the adjacent structural units of the MCLR-BTPs to PP2A by weakening the hydrogen bonds and ionic bonds for some of the key interaction sites. Finally, a specific investigation of the interactions associated with the Mn^2+^ ions (in the catalytic center) found that the weakening combination of the MCLR-BTPs to PP2A further weakened some of the ionic bonds and metal bonds between the conserved amino acids and the Mn^2+^ ions, resulting in the increased exposure of Mn^2+^ ions. At the same time, the weakening of the hydrogen bond Arg^4^←Arg_214_ increased the exposure of Arg_214_; thus, the combination of the phosphate group to Arg_214_ was facilitated. In this way, the catalytic activity of the PP2A was restored. This study comprehensively evaluated the toxicity of the MCLR-BTPs, thereby contributing to the reduction in the secondary environmental risk caused by MCLR-BTPs, improving the use of regulatory strategies against MC/MC-BTP biotoxicity, and minimizing its impact on human health.

## Figures and Tables

**Figure 1 ijerph-20-00964-f001:**
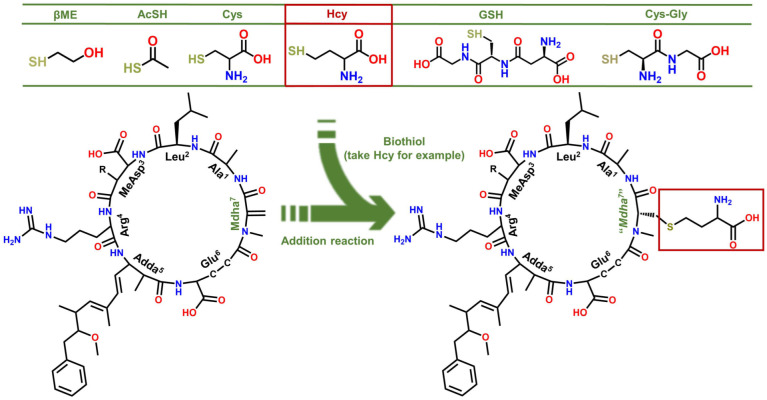
Structures of the MCLR-BTPs resulting from the nucleophilic addition reactions of biothiols to Mdha^7^.

**Figure 2 ijerph-20-00964-f002:**
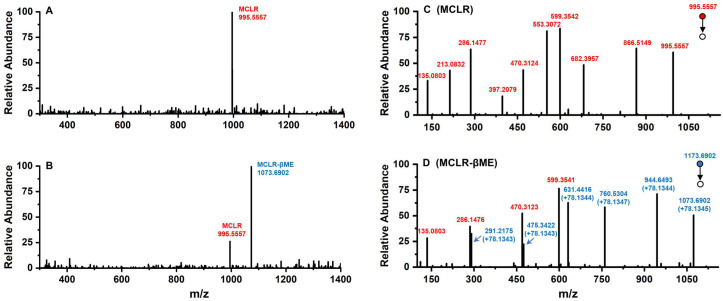
The MS analysis of MCLR and its electrophilic addition product, MCLR-βME. Conditions: MS spectra for MCLR (**A**) and MCLR-βME (**B**); MS/MS spectra for MCLR (**C**) and MCLR-βME (**D**).

**Figure 3 ijerph-20-00964-f003:**
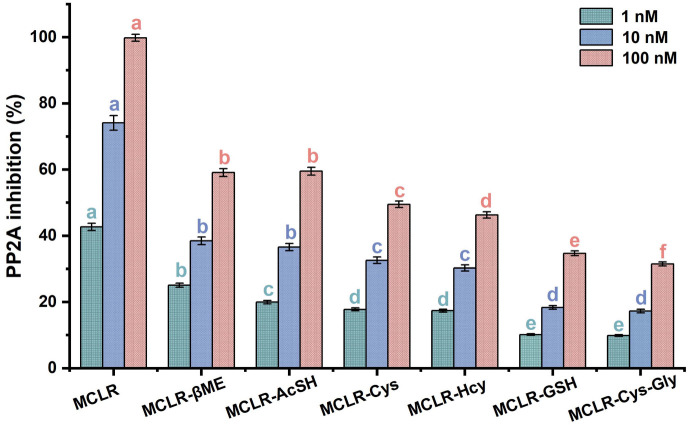
The inhibition effects of MCLR and MCLR-BTPs on PP2A. The error bar is the standard error of three repeated analyses. A one-way analysis of variance (ANOVA) followed by LSD post hoc tests was used to verify significant differences among MCLR and MCLR-BTPs. Different letters indicate significant differences between groups (*p* < 0.05), obtained using **SPSS** software.

**Figure 4 ijerph-20-00964-f004:**
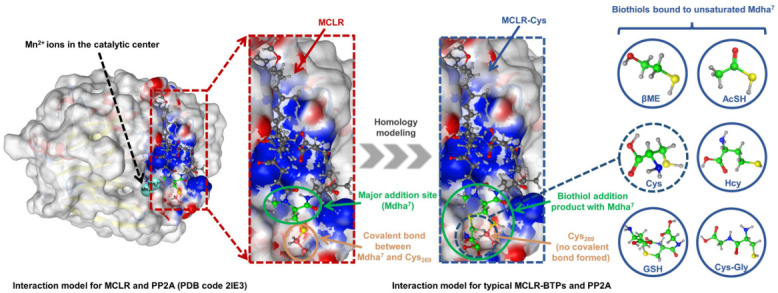
Illustration of interaction-model construction for MCLR-BTP-PP2A complexes (with no PDB models) based on homology modeling strategy.

**Figure 5 ijerph-20-00964-f005:**
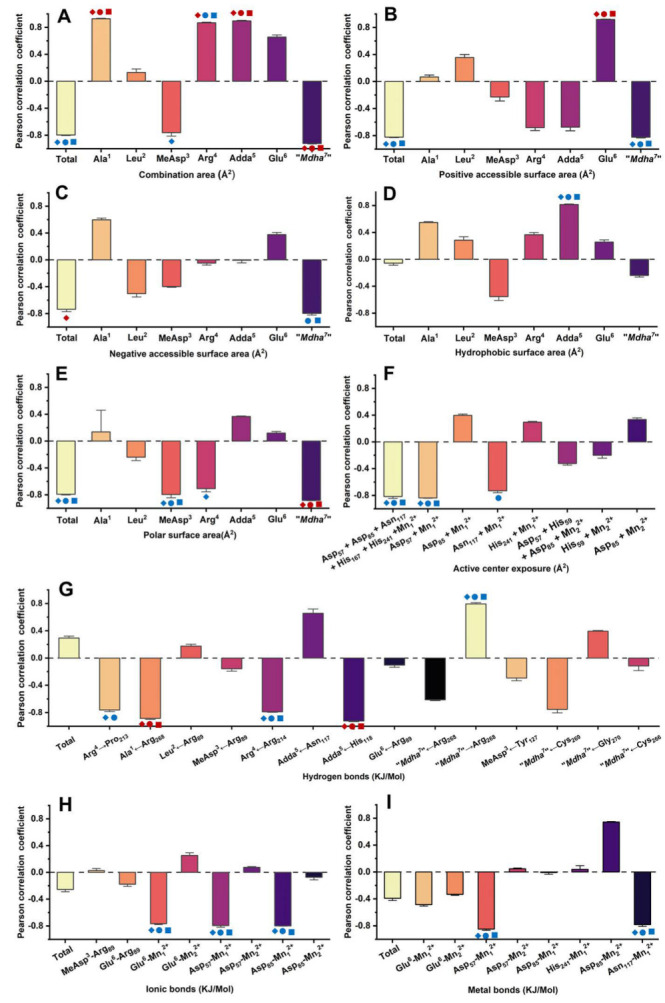
Pearson correlation coefficients between inhibition data and (**A**) combination areas, (**B**) positive accessible surface areas, (**C**) negative accessible surface areas, (**D**) hydrophobic surface areas, (**E**) polar surface areas, (**F**) active center exposure, (**G**) hydrogen bonds, (**H**) ionic bonds, and (**I**) metal bonds. Conditions: The symbols 
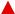
, 
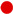
, 
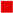
 mean that the interaction parameters are extremely significantly correlated with the inhibition data at the levels of 1, 10, and 100 nM, respectively (*p* < 0.01). The symbols 
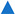
, 
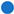
, 
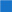
 mean that the interaction parameters are significantly correlated with the inhibition data at the levels of 1, 10, and 100 nM, respectively (*p* < 0.05).

**Figure 6 ijerph-20-00964-f006:**
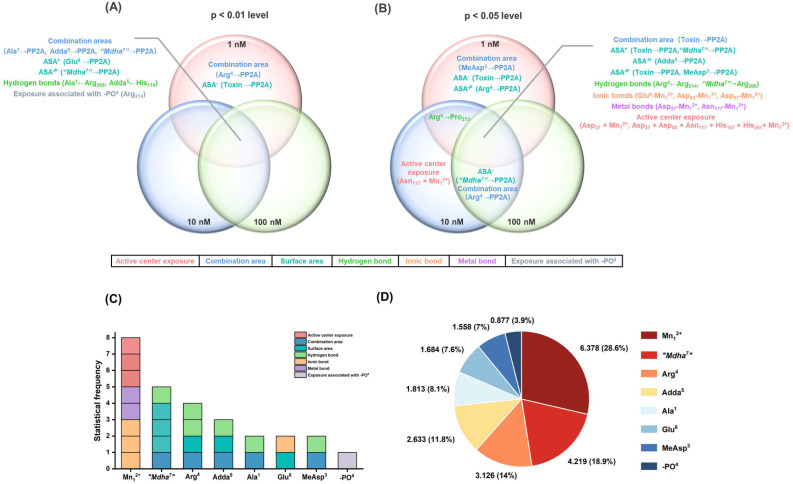
Venn diagrams of the significant interaction parameters at the *p* < 0.01 level (**A**) and *p* < 0.05 level (**B**). Histograms for the statistical frequency (**C**) related to the key interaction sites and pie chart for the total |$\bar{R}$| values (**D**) related to the key interaction sites. Conditions: ASA^+^ stands for positive accessible surface area, ASA^−^ stands for negative accessible surface area, ASA^−H^ stands for hydrophobic surface area, and ASA^−P^ stands for the polar surface area; $\bar{R}$ is the average of Pearson correlations at three toxin concentrations.

**Figure 7 ijerph-20-00964-f007:**
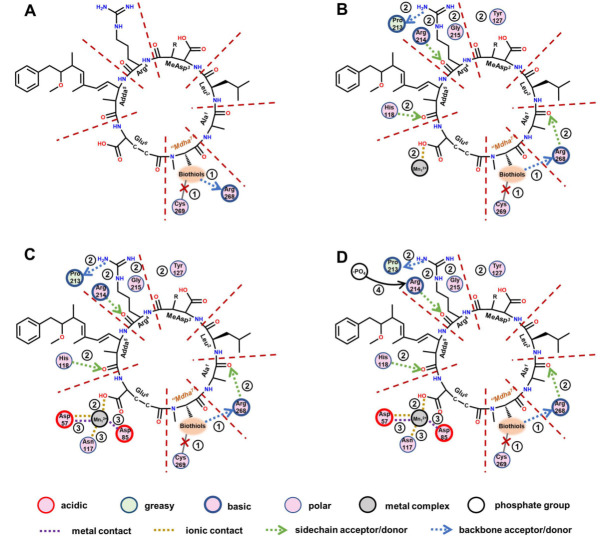
The two-dimensional ligand–receptor interaction diagram for the combination of the toxins to PP2A. The direct influence of the introduced biothiols on the interactions between the toxins and PP2A (**A**). The influence of the changed “Mdha^7^” on the interactions between the toxins and the PP2A (**B**). The influence of the weakening combination of the toxins to PP2A on the interactions involving Mn^2+^ ions (**C**). The influence of the changed interactions on the exposure of amino acids bound to a phosphate group (**D**).

## Data Availability

The authors declare that all data and materials are available to be shared on a formal request.

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
