# Peer review of "Regulation Effectiveness and Mechanism of Biotransformation Pathway on the Toxicity of Microcystin-LR Target to Protein Phosphatase 2A"

_ijerph, 2023, doi:10.3390/ijerph20020964_

Round 1

Reviewer 1 Report

The manuscript is well written but the work importance isn't well defined. In introduction section they write about hepatotoxicity and carcinogenicity, however isn't clear how the microcystins damage the cell and what is the importance of phospatase to mantain the cell ballance. Methods are well describe and could be reproduced. The resulta are clearly presented and disscused, with the exception that isn't describe how these finding would contribute to inhibit the cell damage. The conclusions are very extensive and confuse, the authors should be brief and write some presperpective for describing if this kind of investigation could contribute to enviroment control and health manteinance.

Reviewer 2 Report

This research investigated the regulation effectiveness and mechanism of the biotransformation pathway of the Microcystins.  The chemical and statistical analysis had done enough. The findings are novel. But this manuscript is not written well. The text is poorly arranged hence difficult to read. There is no usable discussion. The authors have presented only the results in the results and discussion section. Make the manuscript readable first. The results and discussion section should be improved and rewritten.  

Round 2

Reviewer 2 Report

The manuscript has improved a lot, especially the discussion. Now, this article is meaningful and valuable for readers.